# Grid-Based Structural and Dimensional Skin Cancer Classification with Self-Featured Optimized Explainable Deep Convolutional Neural Networks

**DOI:** 10.3390/ijms25031546

**Published:** 2024-01-26

**Authors:** Kavita Behara, Ernest Bhero, John Terhile Agee

**Affiliations:** 1Department of Electrical Engineering, Mangosuthu University of Technology, Durban 4031, South Africa; beharak@mut.ac.za; 2Discipline of Electrical, Electronic and Computer Engineering, University of KwaZulu Natal, Durban 4041, South Africa; ageej@ukzn.ac.za

**Keywords:** explainable convolutional neural network, skin cancer, grid-based structural pattern, VGG-16, adaptive intelligent coney optimization algorithm

## Abstract

Skin cancer is a severe and potentially lethal disease, and early detection is critical for successful treatment. Traditional procedures for diagnosing skin cancer are expensive, time-intensive, and necessitate the expertise of a medical practitioner. In recent years, many researchers have developed artificial intelligence (AI) tools, including shallow and deep machine learning-based approaches, to diagnose skin cancer. However, AI-based skin cancer diagnosis faces challenges in complexity, low reproducibility, and explainability. To address these problems, we propose a novel Grid-Based Structural and Dimensional Explainable Deep Convolutional Neural Network for accurate and interpretable skin cancer classification. This model employs adaptive thresholding for extracting the region of interest (ROI), using its dynamic capabilities to enhance the accuracy of identifying cancerous regions. The VGG-16 architecture extracts the hierarchical characteristics of skin lesion images, leveraging its recognized capabilities for deep feature extraction. Our proposed model leverages a grid structure to capture spatial relationships within lesions, while the dimensional features extract relevant information from various image channels. An Adaptive Intelligent Coney Optimization (AICO) algorithm is employed for self-feature selected optimization and fine-tuning the hyperparameters, which dynamically adapts the model architecture to optimize feature extraction and classification. The model was trained and tested using the ISIC dataset of 10,015 dermascope images and the MNIST dataset of 2357 images of malignant and benign oncological diseases. The experimental results demonstrated that the model achieved accuracy and CSI values of 0.96 and 0.97 for TP 80 using the ISIC dataset, which is 17.70% and 16.49% more than lightweight CNN, 20.83% and 19.59% more than DenseNet, 18.75% and 17.53% more than CNN, 6.25% and 6.18% more than Efficient Net-B0, 5.21% and 5.15% over ECNN, 2.08% and 2.06% over COA-CAN, and 5.21% and 5.15% more than ARO-ECNN. Additionally, the AICO self-feature selected ECNN model exhibited minimal FPR and FNR of 0.03 and 0.02, respectively. The model attained a loss of 0.09 for ISIC and 0.18 for the MNIST dataset, indicating that the model proposed in this research outperforms existing techniques. The proposed model improves accuracy, interpretability, and robustness for skin cancer classification, ultimately aiding clinicians in early diagnosis and treatment.

## 1. Introduction

Skin cancer is a prominent form of cancer that significantly impacts patient survival. Early detection of skin cancer margins is essential to prevent cancer progression to advanced stages and reduce cancer-related fatalities [1]. According to the cancer data provided by the American Cancer Society, the mortality rate for skin cancer is 75%, making it a significant global cancer [2]. Moreover, the World Health Organization (WHO) reports that the diagnosis of skin cancers worldwide affects one-third of the world population [3]. Also, skin cancers have been increasing constantly over the past few decades. The medical statistics in 2020 stated that 1.5 million patients were diagnosed with skin cancer, 325,000 were diagnosed with melanoma, and 57,000 deaths were reported worldwide due to this condition [3,4]. In 2022, the United States reported 97,920 melanoma cases [4]. As per the scientific reports, dangerous UV rays from the sun damage skin cells, resulting in melanoma, thereby increasing the chances of skin cancer. Other factors influencing the development of malignant cells include smoking, alcohol, infections, and the surrounding environment [5].

Skin cancer occurs by the development of tumors in the epidermis, which can be classified as either benign or malignant. Among the types of skin cancers, malignant tumors and pre-malignant tumors are dangerous, resulting in the fatality of affected patients [6]. Malignancy leads to the rapid proliferation of malignant cells, impacting nearby tissues [7]. Furthermore, malignant tumors can divide themselves and travel to other body parts through the lymphatic system, which can lead to the formation of a new tumor in another organ. Unlike malignant tumors, benign tumors never invade or affect nearby tissues [7]. When malignant tumors are left untreated, the tumor will spread all over the body, leading to the death of the patient. One of the most commonly prevailing skin cancers is basal cell carcinoma, which, though slower in growth, badly affects the tissues around it and destroys them [8].

For skin cancer screening, dermatologists use macroscopic and dermascopic images with good resolution that visualize the skin features, which assists in diagnosing skin cancer [9]. Furthermore, to support the visual features, MetaNet, a multiplication method that generates coefficients of the skin from a one-dimensional convolution sequence, promotes the accuracy of skin cancer classification due to the finely refined visual features [10]. The accuracy of skin cancer detection using computer vision technology and machine learning methods is worth investigating [11]. Such advanced technologies concentrate on lesion detection, lesion segmentation, and the detection of skin diseases [12]. Moreover, the methods of feature extraction and feature representation support the accuracy of detection [13]. Advances in machine and deep learning methods increase the accuracy of skin cancer detection [14].

Additionally, focusing on the patient’s history and genetic background impacts the detection accuracy and learning from the skin features, promoting accurate detection [15]. Despite these significant advancements and achievements, existing approaches often suffer from challenges. For example, Ahmed Magdy [14] developed two methods, which are a pre-trained deep neural network with K-nearest neighbor (PDNN-KNN) and Alexnet grey wolf optimization (AlexGWO). Both models achieved a very high performance compared to most existing methods. However, the time taken for the classification was very high. Lisheng Wei et al. [16] introduced a lightweight CNN model using segmentation and feature extraction to improve the accuracy of skin cancer classification. However, the model’s limited efficiency stems from its reliance on data; a sizable dataset is required for model training.

A model’s efficacy is diminished if the segmentation image size is excessively huge. Andre et al. [17] implemented a MetaBlock attention-based framework for skin cancer classification that utilizes metadata to improve the feature map extracted from the images. However, metadata only has an advantage in the training phase for data available in the domain. Combining metadata and the CNN model increases the model’s size, increasing the system’s complexity. Shancheng Jiang et al. [18] utilized a DRANet model that leverages an attention mechanism and a deep learning model. This model performed better than the other former approaches with less parameter size and computational complexity. However, data augmentation has increased the computational cost of deep learning models. The attention mechanism utilized in that research raises the model’s complexity and instability, producing errors. 

Azhar Imran et al. [19] employed an ensemble approach for skin cancer that integrates the VGG, ResNet, and Caps Net models, and demonstrated that the ensemble model performed better than the individual learners in decision-making and perceptive issues. However, ensemble methods have limitations, such as large data requirements and computational complexity.

Adekanmi A. Adegun and Serestina Viriri [20] utilized a CAD framework that combines an encoder-decoder segmentation network and the FCN-based DenseNet classification network, which enhanced the model’s performance and identified complex features. However, DenseNet has a complex architecture that increases computational costs and processing times. Krishna Mridha et al. [21] introduced an optimized CNN model with Grad CAM; the model deals with typical skin cancer problems and helps doctors diagnose skin cancer early. However, the interpretability score of the model is low; to increase the interpretability score, additional transformers are needed. Lubna Riaz et al. [22] designed CNN and LBP architecture to extract an image’s high-level features, enhancing the model’s classification accuracy. However, that model requires a large amount of training data; more dimensionality reduction algorithms are needed for better results. Karar Ali et al. [9] modeled an Efficient B0 and transfer learning-enabled CNN model, showcasing better performance than other existing approaches. That model has more complexity and is prone to overfitting. SBXception, a shallower and broader version of the Xception network, was developed by Abid Mehmood et al. [23] for classifying skin cancer. Decreasing the depth and increasing the width of the architecture improved the performance of Xception with high accuracy. However, that model only considers seven different kinds of skin lesions, and there might be problems with generalizability. 

Yonis Gulzar and Sumeer Ahmad Khan [24] provided a detailed comparison research of U-Net and attention-based algorithms for image lesion picture segmentation, which will assist in identifying skin lesions. TransUNet’s hybrid architecture performs better than previous benchmarking techniques in both qualitative and quantitative aspects. The TransUNet technique is resilient to noise and low contrast, but there may be interpretability issues. Sumeer Ahmad Khan et al. [25] developed hand-crafted HSIFT features to classify medical images. In terms of the ability to discriminate between features while displaying category labels, the developed HSIFT feature performs better than CNN’s feature, but there are still some misclassifications. Shahnawaz Ayoub et al. [26] developed a model to augment the dataset using GAN, solving the imbalanced dataset issue. However, the GAN model has difficulties regarding non-convergence and mode collapse. Hardik Nahata and Satya P. Singh [27] developed a CNN model for skin cancer classification. That model shows efficient performance with a reduced cost, but the requirement for computational resources is high. Nadia Smaoui Zghal and Nabil Derbel [28] created an automated approach for detecting pigmented skin lesions. Although that model’s excellent accuracy indicates its dependability, generalizability problems could arise. V. Srividhya et al. [29] created a CNN model for the classification of skin lesions. Despite the method’s compelling performance in classification, misclassification still occurred. Therefore, this research proposes a deep learning model to solve the high computational time issues.

Additionally, traditional CNNs primarily focus on extracting local features, neglecting the crucial spatial relationships within the lesions. Mostly, these models rely on manually designed features, which are time-consuming and subjective and limit the model’s ability to learn complex patterns. Also, the intricate internal mechanisms of deep models frequently remain obscure, impeding trust and hindering clinical application.

We propose a novel Grid-Based Structural and Dimensional (GBSD) Explainable Deep CNN architecture to address these challenges for accurate and interpretable skin cancer classification. A systematic diagram of the proposed framework is shown in Figure 1. GBSD leverages the following key innovations of this study:Develop a novel Grid-Based Structural Feature Extraction approach that captures the spatial relationships between lesion pixels within a grid structure, enabling the model to learn complex patterns and context-aware features. The grid-based structural patterns contain nine gray level values, which reduce the intensity variations and improve the model’s performance.Construct a Dimensional Feature Learning to extract relevant features from different image channels, such as color and texture, enriching the model’s lesion representation and improving discrimination between cancer types.Construct a Self-Featured Optimized Explainability technique that dynamically adjusts the network architecture by selecting the most informative features for each image, leading to a more interpretable model and improved classification accuracy. The self-feature selected ECNN model detects and classifies skin cancer; the model’s hyperparameters are tuned by the novel AICO optimization algorithm, which aims to detect skin cancer accurately.Develop an adaptive intelligent coney optimization algorithm (AICO) by combining the adaptive intelligent hunt characteristics of coyotes with the intelligent survival trait characteristics of coneys to improve convergence speed and enhance classification accuracy.The utilization of the adaptive intelligent coney optimization algorithm enables the self-feature selected ECNN to adjust the classifier parameters effectively. The self-feature selected ECNN leverages the AICO algorithm. It leads to an improved capability of the classifier in detecting skin cancer; the system can handle a wide range of skin cancer manifestations and improve diagnostic accuracy by using the AICO algorithm.

The remaining sections of this manuscript are structured as follows: the experimental results and discussion for the skin cancer classification are presented in Section 2 and Section 3, the AICO self-feature selected ECNN model’s methodology is discussed in Section 4, and Section 5 contains the conclusion and future scope of the research.

## 2. Results

Our proposed novel Grid-Based Structural and Dimensional Deep CNN (GBSD) architecture demonstrated remarkable performance in the classification of skin cancer, outperforming current techniques and providing a crucial understanding of lesion characteristics. This section explains the experimental results obtained from the proposed model for skin cancer classification.

### 2.1. Experimental Setup

This section presents the empirical findings of the proposed AICO self-featured selected ECNN. The proposed model is evaluated at each stage using qualitative and procedural methodologies. The experimental tests in this study were conducted using Python 3.7 on a 4 GHz Intel Core i7 CPU running at a rate of 1.80 GHz, with 2304 MHz, four cores, and eight logical processors. The testing also utilized an NVIDIA K80 GPU with 12 GB of RAM and a speed of 4.1 TFLOPS.

### 2.2. Dataset Description

The experiments were performed on two datasets, HAM10000 and ISIC, obtained from the ISIC repository. The images in the datasets have been classified as benign and malignant.

(a)Skin Care MNIST; HAM1000 [30]: The dataset comprises 10,015 dermascope images, encompassing a comprehensive range of significant diagnostic categories.(b)The skin cancer ISIC dataset [31] comprises 2357 images of malignant and benign oncological diseases from The International Skin Imaging Collaboration (ISIC). The images were categorized based on the ISIC classification, and each subset contains an equal number of images, except for melanomas and moles, which have a slightly higher representation.

### 2.3. Performance Metrics

This study’s skin lesion classification model was evaluated for effectiveness using performance indicators, including accuracy, the critical success index (CSI), false positive rate (FPR), and false negative rate (FNR) [32].

(a)Accuracy

Accuracy is the proportion of correctly identified skin cancer lesions to the total number of predictions made by the model:(1)$$accuracy=\frac{tp+tn}{tp+tn+fp+fn}$$

(b)Critical Success Index (CSI)

CSI is also known as the threat score. It is the verification measure of the total number of correctly predicted skin cancers (hits) to the total number of predictions made by the model, including false predictions:(2)$$CSI=\frac{tp}{tp+fp+fn}$$

(c)False Positive Rate (FPR)

FRP is defined as the proportion of negative cases incorrectly identified as positive cases in the data, which is mathematically represented as:(3)$$FPR=\frac{fp}{fp+tn}$$

(d)False Negative Rate (FNR)

FNR is defined as the proportion of positive cases incorrectly predicted to be negative cases in the data, which is mathematically calculated as:(4)$$FNR=\frac{fn}{tp+fn}$$

### 2.4. Experimental Outcomes

The primary objective of this study was to develop a model for classifying skin cancer using dermascope images. Firstly, skin lesion images were collected from the publicly available datasets MNIST HAM10000 and ISIC, consisting of dermascope images from distinct populations. Next, in the pre-processing phase, adaptive threshold ROI extraction was performed for image quality enhancement. The pre-processed image was fed as input to the feature extraction module, which consisted of a grid-based structural pattern, grid-based directional pattern, statistical features, and VGG16 for extracting the valuable features from the image. The extracted features were fed into the self-feature selected optimized explainable deep convolutional neural network model for skin cancer classification. The model’s performance was optimized by the novel coyote coney optimization algorithm inspired by coyotes’ and coneys’ adaptive and hunting traits. The optimization algorithm fine-tunes the classifier’s hyperparameters and improves the model’s classification accuracy.

The experimental results of the GBSD-EDCNN self-feature selected optimizer are depicted in Figure 2, which illustrates the original image, adaptive region of interest (ROI) extracted image, local binary pattern (LBP), local directional pattern (LDP), VGG 16, statistical feature extracted images, Grad CAM++, Full Grad, and the fusion output images. The LBP and LDP features were extracted with a grid-based structural pattern. Here, the grid-based structural pattern takes every 3 × 3 matrix, which helps classify skin cancer using the skin lesion images. LBP has a high tolerance regarding illumination changes, and LDP is robust in noisy situations. The extraction of statistical features helps accurately classify images since the mean, median, and mode of the images are evaluated, and the images with the same values can be classified as the same class. The features extracted from VGG 16, a deep architecture, are more representative of the image content than those extracted from shallower architectures. So, the extraction of these features helps to improve the image classification of the model.

### 2.5. Feature Extraction Phase Using VGG 16

The feature extraction step is essential for the effectiveness of convolutional neural networks (CNNs) in image classification applications. In this stage, the network can recognize and isolate significant characteristics from the input image, such as edges, textures, and forms. These features are subsequently employed by the subsequent layers for classification. We have performed feature extraction using three pre-trained models, namely, ResNet101, AlexNet, and VGG16. VGG16 performed higher in accuracy, with a level of 0.95, compared to the other models. VGG16 uses deeper convolutional neural networks (CNNs) with stacked layers of small filters (3 × 3). These enable them to extract complex features and achieve high accuracy on tasks like image classification.

VGG16’s deep architecture, while demanding in terms of energy, has proven to be highly effective in image classification tasks. Its accuracy and superior feature extraction capabilities make it a more suitable choice for our specific objective of skin cancer classification. The features extracted by VGG16 capture intricate patterns and subtle characteristics within skin lesion images, contributing to improved classification accuracy.

While more energy-efficient models such as MobileNetV2 and Xception exist, the trade-off between computational efficiency and the representative power of the extracted features led us to choose VGG16. The depth of the architecture allows it to capture complex hierarchical features crucial for accurate skin cancer classification. Table 1 depicts the performance accuracy at the feature extraction phase. All models demonstrated a gradual increase in accuracy as the number of epochs rose, and the models achieved higher accuracy on the MNIST dataset than the ISIC. This is due to MNIST images being more standardized than ISIC images. Table 1 depicts that VGG16 consistently achieved higher accuracy when compared to the other models.

### 2.6. Performance Analysis of AICO Self-Feature Selected ECNN Model with TP

The performance analysis of the AICO self-feature selected ECNN model with TP 80 concerning the accuracy, CSI, FPR, and FNR for the datasets ISIC and MNIST is illustrated in Table 2. Figure 3a,b show the confusion matrixes for the ISIC and MNIST datasets.

For the ISIC dataset, the AICO self-feature selected ECNN model attained an accuracy of 0.91 for epoch 20, 0.92 for epoch 40, 0.94 for epoch 60, 0.95 for epoch 80, and 0.96 for epoch 100. The CSI value attained by the AICO self-feature selected ECNN model for epoch 20 was 0.92, for epoch 40 was 0.93, for epoch 60 was 0.95, for epoch 80 was 0.96, and for epoch 100 was 0.97. The AICO self-feature selected ECNN model attained minimal FPR values such as 0.09, 0.07, 0.06, 0.05, and 0.03 for the respective epochs 20, 40, 60, 80, and 100. The FNR value of the AICO self-feature selected ECNN model for epoch 20 was 0.07, for epoch 40 was 0.06, for epoch 60 was 0.04, for epoch 80 was 0.03, and for epoch 100 was 0.02, as shown in Figure 4.

The AICO self-feature selected ECNN model’s performance assessment using the MNIST dataset attained valuable results, showing that the accuracy of the model for the epochs 20, 40, 60, 80, and 100 were 0.90, 0.92, 0.93, 0.95, and 0.96 respectively. The CSI value attained by the AICO self-feature selected ECNN model for epoch 20 was 0.91, for epoch 40 was 0.93, for epoch 60 was 0.94, for epoch 80 was 0.96, and for epoch 100 was 0.97. For epochs 20, 40, 60, 80, and 100, the AICO self-feature selected ECNN model yielded a minimal FPR value, such as 0.09, 0.08, 0.06, 0.05, and 0.04, respectively. The FNR value of the AICO self-feature selected ECNN model for epoch 20 was 0.08, for epoch 40 was 0.06, for epoch 60 was 0.05, for epoch 80 was 0.03, and for epoch 100 was 0.02, as shown in Figure 5.

### 2.7. Comparative Analysis with the Current State-of-the-Art Methods

Traditional approaches such as EfficientNet-B0 [9], lightweight CNN [16], DenseNet [20], CNN [21], ECNN, COA-ECNN, and ARO-ECNN were used for comparative analysis.

#### 2.7.1. Comparative Analysis with TP for the ISIC Dataset

The comparative assessment of the AICO self-feature selected ECNN with the traditional approaches in terms of accuracy, FPR, and FNR for the ISIC dataset is depicted in Figure 6. The accuracy attained by the AICO self-feature selected ECNN model was 0.96, which was improved by 18.26% over lightweight CNN, 21.51% over DenseNet, 19.52% over CNN, 6.86% over EfficientNet-B0, 5.49% over ECNN, 2.43% over COA-ECNN, and 5.17% over ARO-ECNN. The AICO self-feature selected ECNN model achieved 0.97 CSI, which showed improvements of 16.54%, 19.76%, 17.8%, 6.79%, 5.43%, 2.40%, and 5.12% over the existing methods lightweight CNN, DenseNet, CNN, EfficientNet-B0, ECNN, COA-ECNN, and ARO-ECNN, respectively. The FPR value of the AICO self-feature selected ECNN model was 0.03; when compared with the conventional approaches, the FPR value was reduced due to the increased TP value. Similarly, when compared with traditional approaches such as lightweight CNN, DenseNet, CNN, Efficient Net B0, ECNN, COA-ECNN, ARO-ECNN, and AICO-ECNN, the novel AICO self-feature selected ECNN model attained an FNR value of 0.02 with TP 80, which shows that the proposed model minimizes the FPR and FNR rates with increasing TP values and thus its lower FPR and FNR values enhance the model’s classification accuracy.

#### 2.7.2. Comparative Analysis with K-Fold for the ISIC Dataset

Figure 7 displays the comparative analysis of the AICO self-feature selected ECNN model with the ISIC dataset concerning the accuracy, CSI, FPR, and FNR with k-fold value 10. When compared with the conventional techniques like lightweight CNN, DenseNet, CNN, Efficient Net-B0, ECNN, COA-ECNN, and ARO-ECNN, the accuracy improvement of the AICO self-feature selected ECNN model was 17.65%, 20.46%, 18.79%, 5.98%, 5.8%, and 4.3% respectively. The CSI value attained by the AICO self-feature selected ECNN model was 0.96, which is far better than the conventional techniques. The CSI improvement percentage over lightweight CNN was 16.05%, DenseNet was 18.84%, CNN was 17.18%, Efficient Net-B0 was 5.96%, ECNN was 5.78%, COA-ECNN was 4.28, and ARO-ECNN was 5.69%. The respective FPR and FNR values of the AICO self-feature selected ECNN model were 0.044 and 0.04, which shows that the proposed model produces more minimum FPR and FNR values than the existing research. The minimum FPR and FNR values indicate that the AICO self-feature selected ECNN model accurately classifies skin cancer better than conventional methods.

#### 2.7.3. Comparative Analysis with TP for the MNIST Dataset

Figure 8 shows the comparative analysis of the AICO self-feature selected ECNN model using the MNIST dataset with TP values concerning the accuracy, CSI, FPR, and FNR. For the increasing TP value, the model attained an accuracy of 0.96, which is comparably greater than the existing technique lightweight CNN by 17.4%, DenseNet by 21.13%, CNN by 18.86%, Efficient Net-B0 6.36%, ECNN by 5.81%, COA-ECNN by 2.82%, and ARO-ECNN by 4.24%. The AICO self-feature selected ECNN model with TP 80 achieved 0.97 CSI, which showed improvements of 16.54%, 19.76%, 17.8%, 6.79%, 5.43%, 2.40%, and 5.12% over the existing methods lightweight CNN, DenseNet, CNN, EfficientNet-B0, ECNN, COA-ECNN, and ARO-ECNN, respectively. The FPR and FNR values attained by the AICO self-feature selected ECNN model were 0.4 and 0.2, comparably less than the conventional approaches. The reduced FPR and FNR due to the increased TP demonstrated that the AICO self-feature selected ECNN classification model enhances skin cancer detection accuracy and reliability.

#### 2.7.4. Comparative Analysis with k-Fold for the MNIST Dataset

Figure 9 displays the comparative analysis of the AICO self-feature selected ECNN model with the ISIC dataset concerning the accuracy, CSI, FPR, and FNR with k-fold value 10. The percentage improvement for accuracy with k-fold 10 over the conventional techniques showed that the AICO self-feature selected ECNN improved the accuracy of the classification model. The AICO self-feature selected ECNN attained 0.95 accuracy, which showed an improvement over lightweight CNN, DenseNet, CNN, EfficientNet-B0, ECNN, COA-ECNN, and ARO-ECNN by 17.29%, 20.24%, 18.79%, 5.76%, 4.28%, 4.12%, and 4.28%, respectively. The CSI value attained by the AICO self-feature selected ECNN model was 0.95, which is far better than the conventional techniques. The CSI improvement percentage over lightweight CNN was 15.67%, DenseNet was 18.62%, CNN was 17.08%, Efficient Net-B0 was 5.74%, ECNN was 4.27%, COA-ECNN was 4.11, and ARO-ECNN was 4.27%. The respective FPR and FNR values of the AICO self-feature selected ECNN model were 0.044 and 0.048, which shows that the proposed model produces more minimum FPR and FNR values than the existing methods.

### 2.8. Ablation Study

#### 2.8.1. Ablation Study on VGG-16 Model with ISIC and MNIST Dataset

Figure 10 depicts the ablation study of the pre-trained VGG-16 model with other feature extraction techniques for accuracy with the ISIC and MNIST datasets. For the ISIC dataset, the accuracy rate attained by ECNN with VGG 16 was 0.91 with TP 80 when compared with the conventional ResNet 101 and AlexNet models. The VGG 16 feature extraction models improved by 9.24% and 15.00%, respectively. Similarly, for the MNIST dataset, the ECNN with VGG 16 model achieved a 0.90 accuracy rate, which showed a 7.66% improvement over ResNet 101 and 15.01% over the AlexNet feature extraction model.

#### 2.8.2. Ablation Study on the AICO Self-Feature Selected ECNN with and without Feature Extraction

Figure 11 depicts the ablation study on the AICO self-feature selected ECNN with and without feature extraction in terms of accuracy for the ISIC and MNIST datasets. For the ISIC dataset, the accuracy attained by the proposed model with feature extraction was 0.96 with TP 80. When compared with the AICO self-feature selected ECNN without feature extraction, the proposed model with feature extraction obtained an improvement of 5.35%. Similarly, for the MNIST dataset, the AICO self-feature selected ECNN with feature extraction achieved a 0.96 accuracy, showing a 4.41% improvement over the AICO self-feature selected ECNN without a feature extraction model.

### 2.9. Time Complexity Analysis

Figure 12 depicts the time complexity analysis of the AICO self-feature selected ECNN with existing methods like KNN-PDNN [14], lightweight CNN [16], AlexGWO [14], DenseNet [20], CNN [21], EfficientNet-B0 [9], ECNN, COA-ECNN, and ARO-ECNN. Figure 12 shows the proposed model was substantially faster than all other comparative methods, with the lowest computational time of 0.55 s at the 100th iteration. The other existing methods take a computational time of 0.99 s for KNN-PDNN [14], 0.99 s for lightweight CNN [16], 0.94 s for AlexGWO [14], 0.87 s for DenseNet [20], 0.80 s for CNN [21], 0.78 s for EfficientNet-B0 [9], 0.78 s for ECNN, 0.58 s for COA-ECNN, and 0.56 s for ARO-ECNN for the 100th iteration, which requires more time than the proposed method. Even though KNN-PDNN [14] and AlexGWO [14] attain a higher performance than the proposed method, these models require more computational time than the proposed method. So, it can be concluded that the proposed AICO self-feature selected ECNN method is more efficient than most existing methods in terms of both performance and computational time.

## 3. Discussion

The existing techniques employed for skin cancer classification, such as lightweight CNN, DenseNet, CNN, EfficientNet-B0, and ECNN, have some limitations. Errors in the datasets affect the reliability of the model. In lightweight CNN, integrating CNN with metadata increases the size and system complexity. EfficientNet-B0 model overfitting issues impact classification accuracy. The DenseNet model may produce lower interpretability scores, limiting the performance of conventional approaches in skin cancer classification tasks. In response to these shortcomings, the current study presented an AICO self-feature selected ECNN model that achieves better accuracy while avoiding the abovementioned issues. The use of AICO in the model helps to improve the performance of the model by tuning the parameters of the classifier, and the self-feature selected ECNN used in the model solves the overfitting issue because of the incorporation of Grad-CAM++, which does not require re-training or architectural changes for its visual explanation. As a result of fast convergence in AICO, the model helps attain effective results with low computational time, which solves the complexity issues in the model. Table 3 depicts the comparative discussion of the AICO self-feature selected ECNN model with the traditional approaches.

## 4. Materials and Methods

This study proposes a novel approach for skin lesion classification using a grid-based structural and dimensional AICO self-feature selected ECNN. The methodology for this study is discussed in the subsequent sections below.

### 4.1. AICO Self-Feature Selected ECNN

Skin cancer is a prominent cancer worldwide; early detection of skin cancer margins is essential to prevent cancer progression to advanced stages and reduce cancer-related fatalities. The former approaches employed for skin cancer have some limitations, such as system complexity and data requirements. The feature extraction and selection methods utilized in the existing research rely on hand-crafted features and do not extract high-level features from the images. To overcome these issues, the AICO self-feature selected ECNN was developed in this research. In this research, the skin cancer images were initially collected from the publicly available datasets of skin cancer, the MNIST HAM10000 [30] and ISIC [31] datasets. The skin cancer data were initially exposed to the pre-processing stage, and Adaptive Thresholding ROI Extraction was performed for image quality enhancement. The pre-processed image was fed as input to the feature extraction module, which consisted of a grid-based structural pattern, grid-based directional pattern, statistical features, and VGG16 for extracting the valuable features from the image. The extracted features were fed into the self-feature selected optimized explainable CNN model for skin cancer classification. The model’s performance was optimized by the novel coyote coney optimization algorithm inspired by coyotes’ and coneys’ adaptive and hunting traits. The optimization algorithm fine-tunes the classifier’s hyperparameters and improves the model’s classification accuracy. A schematic diagram of the proposed framework is shown in Figure 1.

### 4.2. Image Input

The images collected from the ISIC and MNIST datasets were fed as input to the AICO self-feature selected ECNN model, which is mathematically represented as:(5)$$Q=\left\{I_{1}\ldots \ldots \ldots \ldots I_{z}\right\}$$
where Q represents the dataset and $\left\{I_{1}\ldots \ldots \ldots \ldots I_{z}\right\}$ represents the total number of images in the dataset from 1 to z.

### 4.3. Pre-Processing: Adaptive Thresholding-Based ROI Extraction

Pre-processing was applied to the input images to remove their noise and artifacts. Here, a morphological filter and a blur filter were used for pre-processing. The morphological filter removes the hair artifacts in the dermascope images, and the blur filter is used to smooth the images and their surfaces. Thus, using these filters, the noise and artifacts of the images were removed.

An ROI-based approach was used as a pre-processing step to locate the lesion area in the images accurately. In skin cancer classification, adaptive thresholding can separate the affected regions from the surrounding healthy skin tissue in dermascope images. The adaptive thresholding mechanism utilizes a local threshold value for each pixel in the image based on neighborhood characteristics, allowing better ROI extraction. For each pixel in the image, a local neighborhood or window around that pixel is defined. The size of this window can vary based on the application and the expected size of the objects to be segmented [33]. The pixel’s mean (average) intensity value is calculated within each local window. The pre-processed image is represented as $\;I^{*}$.

### 4.4. Feature Extraction

Feature extraction is a critical step in skin cancer classification from dermoscopy images that involves selecting relevant features from the images that can be used as input for the self-feature selected optimized Explainable CNN model. The choice of features can significantly impact the accuracy and effectiveness of skin cancer classification.

#### 4.4.1. Grid-Based Structural Pattern-LBP Shape-Based Descriptors

Local binary pattern (LBP) is one of the most powerful texture descriptors. It is calculated by comparing the gray level values of the central and the local neighborhood pictures [34]. The LBP operator is described as 3×3 window, and the central pixel of the window is taken as a threshold. The difference between the central pixel and the neighborhood pixels is calculated, and then the neighbor pixels’ values are assigned to 0 or 1 based on the difference. A central pixel in the 3×3 block contains eight neighboring pixels [35]. The LBP descriptor can be calculated as [36,37]:(6)$$LBP_{S,D}({I^{*}}_{o})=\sum_{d=0}^{D-1}U\left(I_{d}^{*}-I_{o}^{*}\right)2^{d}$$
where $I_{o}^{*}=I_{d}^{*}\left(u,v\right)$ is the central pixel of an image $I_{d}^{*}$ at position (u,v) and $I_{d}^{*}=I^{*}\left(u_{d},v_{d}\right)$ represents the neighboring pixel surrounding $I_{o}^{*}$.
(7)$$U\left(t\right)=\left\{\begin{array}{c} 1,ift\geq 0\\ 0,otherwise \end{array}\right.$$
(8)$$u_{d}=u+D\cos\left(2\pi \frac{d}{D}\right)$$
(9)$$v_{d}=v-D\cos\left(2\pi \frac{d}{D}\right)$$
where D denotes the number of neighboring pixels $I_{d}^{*}$ and S  represents the distance between $I_{d}^{*}$ and $I_{o}^{*}$. In traditional LBP descriptors, only the central pixel values are compared with neighboring pixels, which leads to intensity variations. To address the intensity variation, this research introduced a feature extraction model that calculates a gray level value by considering each pixel as a central pixel, which provides nine gray level values. Finally, the mean value is calculated for all gray values, improving the model’s performance.

#### 4.4.2. Grid-Based Directional Pattern-Local Directional Pattern

The local directional pattern (LDP) feature generates an eight-bit binary code for each pixel in an input image. This code is computed by comparing the pixel’s relative edge response values in different directions using edge detectors like Kirsch, Prewitt, or Sobel [36]. The Kirsch edge detector is remarkably accurate, considering responses in all eight neighboring directions. For a central pixel, the eight directional response values $N_{i}=\left(0,\;1,\;2\ldots \ldots 7\right)$ are calculated using Kirsch masks in eight orientations. Not all response values in different directions are equally significant; high responses often indicate the presence of corners or edges in particular directions [37]. To create the LDP, the top $m^{th}$ directional bit responses $B_{i}$ are set to 1, while the remaining bits of the 8-bit LDP pattern are set to 0 [38]. The LDP value of pixel orientation $\left(p_{c},q_{c}\right)$ with different directional responses is given as:(10)$$LDP=\sum_{n=0}^{7}B\left(N_{n}-N_{m}\right)\cdot 2^{n}$$
where B is the binary word, which is represented as:(11)$$B(c)=\left\{\begin{array}{c} 1,ifc\geq 0\\ 0,\;otherwise \end{array}\right.$$
where $N_{m}$ represents the mth significant directional response and $N_{n}$ denotes the response of the Kirsch mask. Here, the grid-based structural pattern takes every 3 × 3 matrix, which helps classify skin cancer using the skin lesion images.

#### 4.4.3. Statistical Features

In image processing, statistical features are quantitative measurements calculated from the pixel intensities in an image [39].

(a)Mean: The mean represents the average intensity value of the pixels within an image.
(12)$$\mu_{Sc}=\sum_{u=1}^{g}\sum_{v=1}^{h}\frac{J(u,v)}{gh}$$
where J(u,v) denotes the pixel’s intensity value at position (u,v) and the image is g by h size.(b)Median ($M_{med}$): The median is a statistical measure that represents the mid-value in a dataset when the data are organized in ascending or descending order. When there is an even number of values in the dataset, the median is calculated as the average of the two middle values.(c)Mode ($M_{mod}$): mode is defined as the value that occurs in a pixel the maximum number of times.

The statistical features are concatenated as $F_{s}=\left[\mu_{Sc}||M_{med}||M_{mod}[]\right]$. The feature dimension of the statistical features is denoted as $\left(1\times 128\times 128\right)$.

#### 4.4.4. VGG 16

The VGG 16 feature extraction model can extract large amounts of data from the image, resulting in better accuracy. The VGG architecture is a small feature extraction model consisting of convolutional, pooling, and fully connected layers [40], as illustrated in Figure 13. In skin cancer image analysis, the VGG16 model is a deep CNN with 16 layers and is renowned for its performance. VGG16 utilizes a 224×224 input image and is divided into five blocks. The initial blocks include 3×3 convolutional layers and 2×2 max pooling layers with 62 and 128 filters, respectively. The subsequent blocks consist of three convolutional layers with 256, 512, and 512 filters, followed by a 2×2 max pooling layer [41]. Even though VGG 16 is known for its high energy consumption, the VGG 16 model is more suitable for feature extraction in image classification tasks than other efficient methods because of its ability to extract high-level features, which leads to high accuracy. The features extracted from VGG 16, a deep architecture, are more representative of the image content than those extracted from shallower architectures.

The LDP, LBP, statistical, and VGG 16 features are concatenated ${I_{f}}^{*}$ and fed into the self-feature selected optimized explainable CNN model.

### 4.5. Self-Feature Selected Optimized Explainable CNN

The self-feature selected explainable CNN is designed to visualize the decisions made by the CNN model, which makes them more explainable and transparent. The traditional CNN techniques have limitations, such as overfitting issues and increasing error probability, and the hybrid approaches enhance system complexity. To overcome these limitations, this research leverages the self-feature selected optimized Explainable CNN model, which incorporates the Grad CAM++ and Full Grad modules. The class discriminative localization technique known as gradient-weighted class activation mapping (Grad CAM++) generates visual explanations for any CNN network without requiring architectural changes or re-training, and the Full Grad calculates the gradient of the biases and sums up [42].

Initially, the input image with the dimension $\left[1\times 128\times 128\right]$ is fed into the explainable CNN model, which consists of Grad CAM++ and Full Grad. The architecture of the explainable CNN model is shown in Figure 8. The Grad CAM++ module produces the Grad CAM image and the guided propagation image, which are used to produce the guided propagation (GP) CAM image. Similarly, the Full Grad module generates two images known as Full Grad and GP, which produces the guided progression Full Grad image [42]. The images generated from the CAM++ and Full Grad modules are fused using Equations (13) and (14).

In Grad CAM++, the saliency map for the given image is calculated using the following Equation [42]:(13)$$H_{i,j}^{q}=relu\left(\sum_{l}w_{l}^{q}.A_{i,j}^{l}\right)$$
where the saliency map for Grad CAM++ is represented as $H^{q}$, and they are the iterators over the pixels in the map, $A^{l}$ represents the visualization of lth feature map and q represents the class. The guided propagation represents the saliency map and the pixel space visualization generated by the GP.

The saliency map from the Full Grad generated image is calculated as [42]:(14)$$P\left(I^{*}\right)=\left\{\begin{array}{l} w_{o}^{T}I_{f}^{*}+bs_{0}\;\;\;\;\;I_{f}^{*}\in Y_{0}\\ \;\vdots \;\;\;\;\;\;\;\;\vdots \\ w_{n}^{T}I_{f}^{*}+bs_{n}\;\;\;\;\;I_{f}^{*}\in Y_{n} \end{array}\right.$$
where $P\left(I_{f}^{*}\right)$ represents the saliency map for the input image $\left(I_{f}^{*}\right)$ from Full Grad, the weights are denoted as $w^{T}$, and the bias terms are denoted as bs.

Thus, the guided propagation of Grad CAM++ and Full Grad [43] is concatenated using Equation (15):(15)$$O_{H}=\frac{1}{2}\left(H_{i,j}^{q}+P\left(I^{*}\right)\right)$$
(16)$$O_{H}=\left\{\begin{array}{l} \frac{1}{2}(relu\left(\sum_{l}w_{l}^{q}.A_{i,j}^{l}\right)+w_{o}^{T}I_{f}^{*}+bs_{0}\;\;\;I_{f}^{*}\in Y_{0}\\ \;\vdots \;\;\;\;\;\;\;\;\;\vdots \\ \frac{1}{2}(relu\left(\sum_{l}w_{l}^{q}.A_{i,j}^{l}\right)w_{n}^{T}I_{f}^{*}+bs_{n}\;\;\;\;\;I_{f}^{*}\in Y_{n} \end{array}\right.$$

The fused image $O_{H}$ with the dimension of $\left[1\times 128\times 128\right]$ is provided as input for the convolutional layers, followed by the max-pooling layers of the CNN model. The convolutional layers extract the meaningful and prominent features from the input image, which generates convolved maps; these maps are fed forward into the ReLU activation function, which generates rectified feature maps. The pooling layers reduce the dimensionality of the feature maps. This study utilized three pooling layers in the explainable CNN model. The dropout layer reduces the overfitting issues by disregarding the nodes present in the layer. The fully connected (FC) layer produces the logistics vector, and the SoftMax function classifies the logistics into the probability of having skin cancer. The tunable parameters, such as weights and biases, are optimized by the AICO algorithm, enhancing the classification accuracy and producing high-quality solutions. Figure 14 depicts the architecture of the self-featured selected ECNN.

### 4.6. Adaptive Intelligent Coney Optimization Algorithm

The AICO algorithm is inspired by the survival traits of coneys and the adaptive and social behavior of coyotes. The algorithm fine-tunes the classifier’s hyperparameters, enhancing the classification accuracy of the Explainable CNN model [44]. Integrating the survival and social behavior of nature-inspired algorithms allows models to prioritize the essential features and reduce the system complexity.

Coneys, also called rabbits [45], are members of the Leporid family. The AICO algorithm considers that rabbits build burrows around their nests and use a random hiding method to hide from predators and hunters. These natural survival strategies are the source of inspiration. To avoid being discovered by predators, these animals avoid proximity to their nests and feed primarily on vegetation such as grass and leafy weeds. We call this kind of foraging approach “exploration.”

Moreover, coneys build several nests and select one randomly for protection to reduce the chance of predators finding them. Exploitation is the term used to describe this random selection technique. Because of their lower trophic level, these animals must alternate between exploration and exploitation depending on their energy situation. The coyote species *Canis latrans* is a North American mammal [46]. The algorithm considers how coyotes organize their social structure and adapt to their surroundings, emphasizing how they share experiences and work together to assault their prey [47]. The AICO algorithm fine-tunes the classifier parameters, such as weights and biases, which enhance the detection and classification accuracy of the skin cancer classification model [48].

#### 4.6.1. Solution Initialization

Based on fitness, the best solution is assigned for predation. In this case, the solution is set up as $R^{t+1}$.

#### 4.6.2. Fitness Evaluation

A higher fitness value denotes a better solution in terms of the goals being pursued. On the other hand, lower fitness values indicate subpar performance. The fitness function for this algorithm is:(17)$$Ft\left(R^{t+1}\right)=accuracy\left(R^{t+1}\right)$$

#### 4.6.3. Primary Predation Phase

In the primary predation phase, the predator searches for the solution by using the following equation:(18)$$R^{t+1}=R^{t}+r_{1}\left(R_{g}^{t}-R_{t}\right)+r_{2}\left(R_{mean}^{t}-R_{t}\right)$$
where $R_{g}^{t}$ represents the global best solution and $R_{mean}^{t}$ denotes the average value.

##### Deviated Search Phase: $\left(\rho_{s}>0.5\right)$

In this phase, the search agent searches beyond the search space if the probability of searching for a solution is greater than 0.5. To avoid searching in the local area, it starts to search for the location of features in all possible directions, so it makes a deviated search toward the search spaces of every other solution:(19)$$R^{t+1}=R^{t}+P(R^{t}-R_{mean})+round\left(0.5\left(0.05+r_{3}\right)\right).n_{1}+V^{t+1}$$

$\rho_{s}=ay$, where y represents the probability of having environmental barriers $y\in \left[0,1\right]$ and a denotes the constant number.
(20)P=L.C
(21)$$L=\left(e-e^{{\left(\frac{t-1}{t_{\max}}\right)}^{2}}\right).\sin\left(2\pi r_{4}\right)$$
(22)$$C=\left\{\begin{array}{l} 1\;\;\;\;\;\;\;\;if\;k==x\left(l\right)\\ 0\;\;\;\;\;\;\;\;otherwise \end{array}\right.$$
where L denotes running length, which ranges from 0 to 1, represents the rounding to the nearest integer, and is the dimension size of the specific problem.
(23)$$x=randperm\left(d\right)$$
where x is the random permutation of the integers from 1 to d, and the search space is represented as d.
(24)k=1,……,d

The next iteration’s velocity is updated using Equation (25):(25)$$V^{t+1}=V^{t}+r_{7}\left|R^{t+1}-R^{t}\right|\;\;\;\;\;\;\;\;\;\;\;\;\;\;r_{7}\in \left[0,1\right]$$
where $V^{t}$ represents the previously followed velocity during the search and $\left|R^{t+1}-R^{t}\right|$ represents distance.

##### Stashing Phase

After finding the best position for acquiring a solution, the search agent obtains the solution from the selected search space. Depending on the value of the energy factor, the predator changes between a deviated search or random hiding. The energy factor is decreased with the increase in iterations, which can strengthen each individual in the population to switch between detour foraging behavior and random hiding behavior:(26)$$R^{t+1}=R^{t}+P\left(r_{6}\left(R^{t}+M.x\left(k\right)R^{t}\right)-R^{t}\right)$$
(27)$$x\left(k\right)=\left\{\begin{array}{l} 1\;\;\;\;\;\;\;\;\;\;\;\;if\;k=\left[r_{7},d\right]\\ 0\;\;\;\;\;\;\;\;\;\;\;else \end{array}\right.$$
where M denotes the hiding parameter and the random numbers are depicted as:(28)$$\left.\begin{array}{l} r_{1}=b_{1}*t_{\max}^{2}*\sin\left(\pi t\right)\\ r_{2}=b_{1}*{\left(1-t\right)}^{2}*\sin\left(\pi t\right) \end{array}\right\}\;\;\;b\in \left(0,4\right)$$
where $\;\;b_{2}\in \left(0,1\right)$, which is represented as a velocity improving parameter, the hiding parameter is described as follows:(29)$$M=\frac{t_{\max}-t+1}{t_{\max}}.r_{6}$$

The random number $r_{6}$ is denoted as $r_{6}=\left\{\begin{array}{l} 1\;\;\;\;\;\;\;\;\;\;if\;\rho_{p}>0.5\\ 0\;\;\;\;\;\;\;\;\;\;else \end{array}\right.$, where the probability of the solutions potion is denoted as $\;\rho_{p}$, which is described as:(30)$$\rho_{p}=\frac{R^{t}-R_{\min}^{t}}{R_{\max}^{t}-R_{\min}^{t}}$$
where $R^{t}$ represents the position at tth iteration, the maximum and minimum position of tth iteration is denoted as $R_{\max}^{t}$ and $R_{\min}^{t}$ respectively, and $r_{5},r_{7}$ are the random numbers 1 or 2.
(31)$$R^{t+1}=R^{t}+P\left(r_{6}\left(R^{t}+M.x\left(k\right)R^{t}\right)-R^{t}\right)$$
(32)$$R^{t+1}=R^{t}\left(1+P\left(r_{6}\left(1+M.x\left(k\right)\right)-1\right)\right)$$

The Taylor series equation is written as enhanced TS, and the updated solution is calculated as:(33)$$R^{t+1}=0.5R^{t}+1.359\;R_{t-1}-1.359\;R_{t-2}\;+0.6795\;\;R_{t-3}$$

Equate the Equations (32) and (33):(34)$$0.5R^{t}+1.359\left(\;R_{t-1}-\;R_{t-2}\right)\;+0.6795\;\;R_{t-3}=R^{t}\left(1+P\left(r_{6}+M.x\left(k\right)r_{6}-1\right)\right)$$
(35)$$R^{t}\left(0.5+P\left(r_{6}+M.x\left(k\right).r_{6}-1\right)\right)=R^{t}\left(1+P\left(r_{6}+M.x\left(k\right)r_{6}-1\right)\right)$$
(36)$$R^{t}=\frac{1}{P_{r}M.x\left(k\right)}\left(1.359\;\left(\;R_{t-1}-\;R_{t-2}\right)\;+0.6795\;\;R_{t-3}\right)$$
(37)$$R^{t+1}=R^{t}\left(1-r_{1}-r_{2}\right)+r_{1}R_{global}^{t}+r_{2}R_{mean}^{t}$$

Substitute (36) in (37), then the equation for a position update is calculated as:(38)$$R^{t+1}=\left[\frac{1}{P_{r}M.x\left(k\right)}\left(1.359\;\left(\;R_{t-1}-\;R_{t-2}\right)\;+0.6795\;\;R_{t-3}\right)\right]\left(1-r_{1}-r_{2}\right)+r_{1}R_{global}^{t}+r_{2}R_{mean}$$

Assume $P_{r}M.x\left(k\right)$ for $\left(0.5+P\left(r_{6}+M.x\left(k\right).r_{6}-1\right)\right)\;$.

Consider that $R_{mean}^{t}$ is the current position of the solution at tth iteration, which is also represented as $R_{mean}^{t}=R^{t}$. The best solution is updated using the following equation:(39)$$R^{t+1}=\left[\frac{1}{P_{r}M.x\left(k\right)}\left(1.359\;\left(\;R_{t-1}-\;R_{t-2}\right)\;+0.6795\;\;R_{t-3}\right)\right]\left(1-r_{1}-r_{2}\right)+r_{1}R_{global}^{t}+r_{2}R^{t}$$

Thus, the optimization algorithm increases the convergence because the fittest search agent makes the search and stashing phases step-by-step. The algorithm enhances the skin cancer classification ability of the explainable CNN model. The flowchart of the algorithm is depicted in Figure 15.

## 5. Conclusions

Skin cancer continues to be a significant worldwide health issue. The AICO self-feature selected ECNN model shows potential for improving the diagnostic precision of the skin cancer classification model. The proposed approach combines the capabilities of the AICO algorithm and the self-feature selected ECNN, resulting in improved efficiency. The ROI extraction approach exhibits exceptional efficacy in detecting the malignant area and improving lesion localization. Utilizing a grid-based structural pattern and statistical features reduces redundancy and aids the models in prioritizing significant features. Incorporating the AICO algorithm optimizes the hyperparameters of the explainable CNN model, resulting in exceptional performance in the classification of skin cancer. This study signifies significant progress in computer-assisted dermatological diagnostics, providing an essential tool for timely skin cancer identification. The model’s performance on the ISIC dataset with TP 80 was evaluated based on accuracy, CSI, FPR, and FNR. The accuracy was measured at 0.96, CSI at 0.97, FPR at 0.03, and FNR at 0.02. The model attained a low computational time of 0.55 s, much less than other existing methods.

Even though the model attains higher accuracy than most existing methods, this high performance is only limited to specific classes in the dataset. If the model is tested under real-time data or any other dataset, there may be deviations in the model’s performance.

In addition, more advancements in deep learning algorithms will be made to enhance the therapeutic outcomes and provide clear explanations, hence promoting trust and acceptance of advanced technologies in medical practice. In future works, we aim to focus on testing this model on real datasets with varying quality, lesion types, and ethnicities to ensure accurate diagnosis in practice and explore interactive explainable frameworks that can foster trust and understanding.

## Figures and Tables

**Figure 1 ijms-25-01546-f001:**
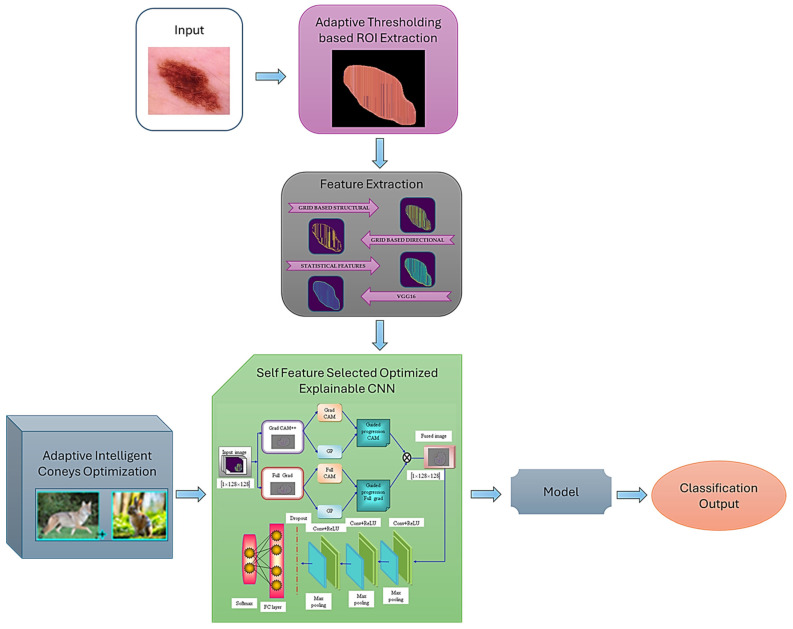
Schematic representation of the AICO self-feature selected optimized ECNN.

**Figure 2 ijms-25-01546-f002:**
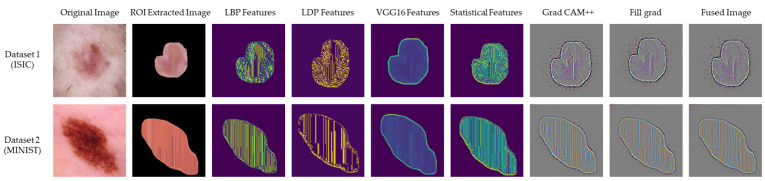
Experimental outcomes of the GBSD-EDCNN skin cancer classification model.

**Figure 3 ijms-25-01546-f003:**
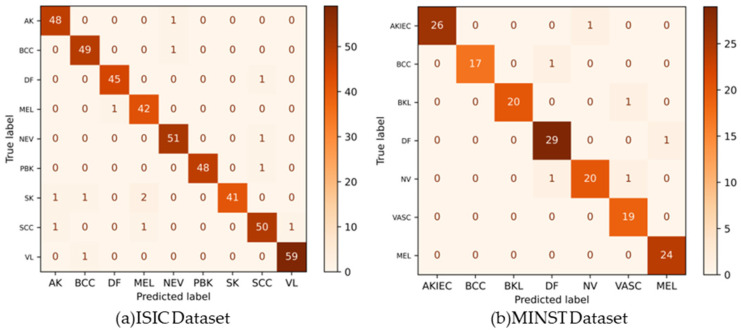
Confusion matrixes for the ISIC and MNIST datasets.

**Figure 4 ijms-25-01546-f004:**
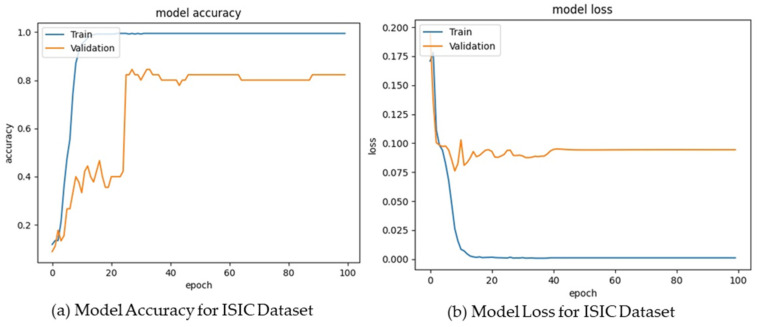
Model Accuracy and Loss for the ISIC Dataset.

**Figure 5 ijms-25-01546-f005:**
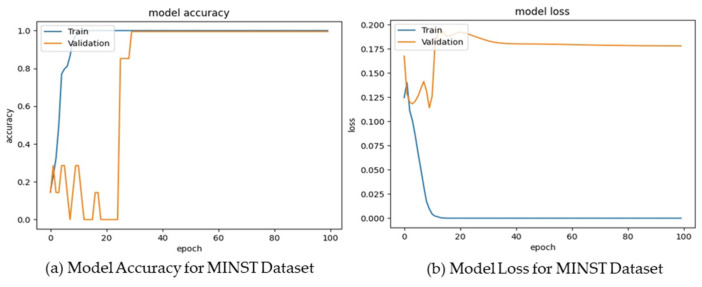
Model Accuracy and Loss for the MNIST Dataset.

**Figure 6 ijms-25-01546-f006:**
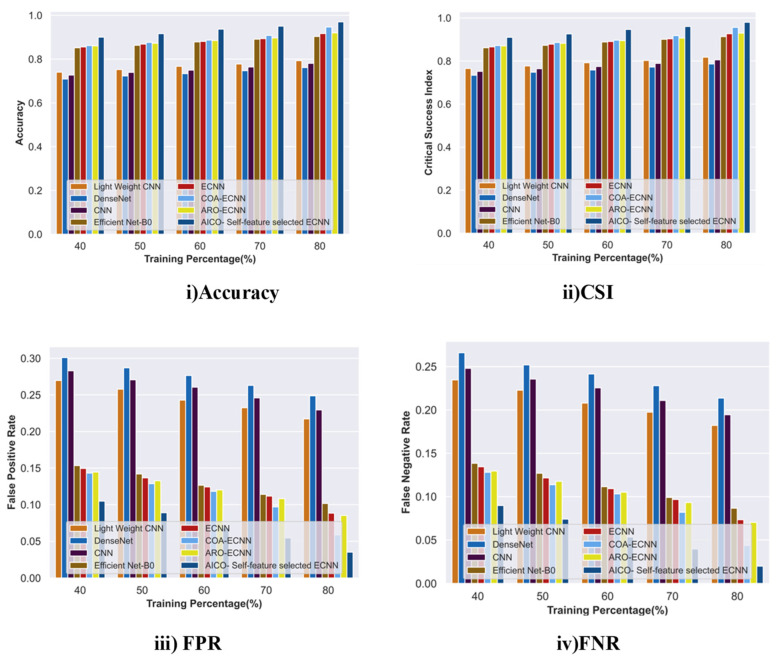
Comparative Analysis with TP for the ISIC dataset.

**Figure 7 ijms-25-01546-f007:**
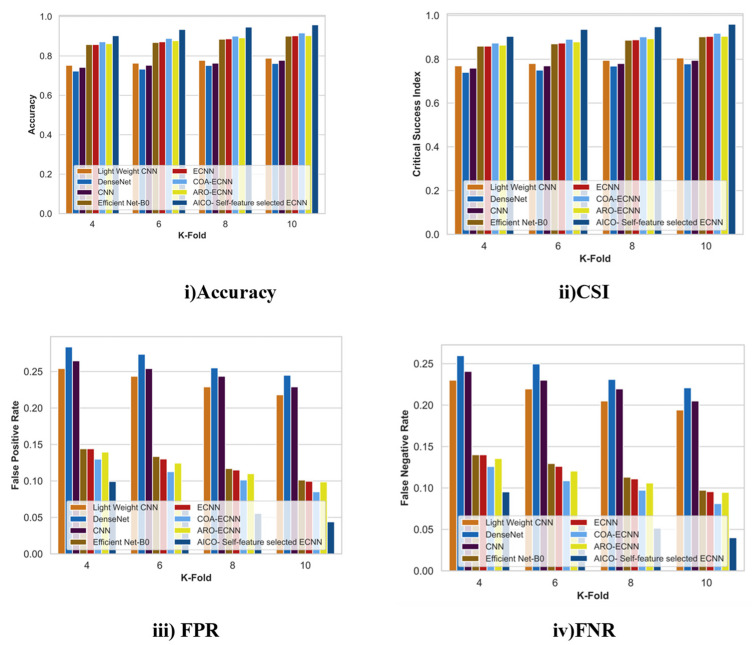
Comparative Analysis with K-fold for the ISIC dataset.

**Figure 8 ijms-25-01546-f008:**
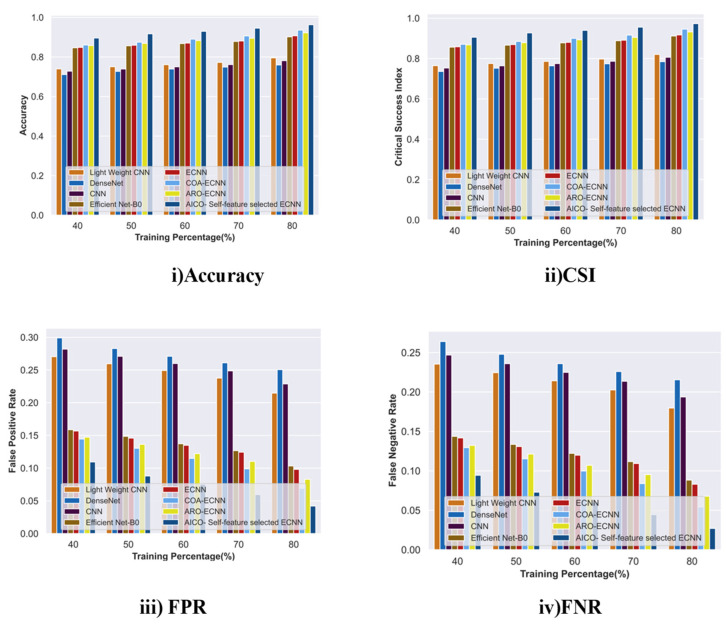
Comparative Analysis with TP for the MNIST dataset.

**Figure 9 ijms-25-01546-f009:**
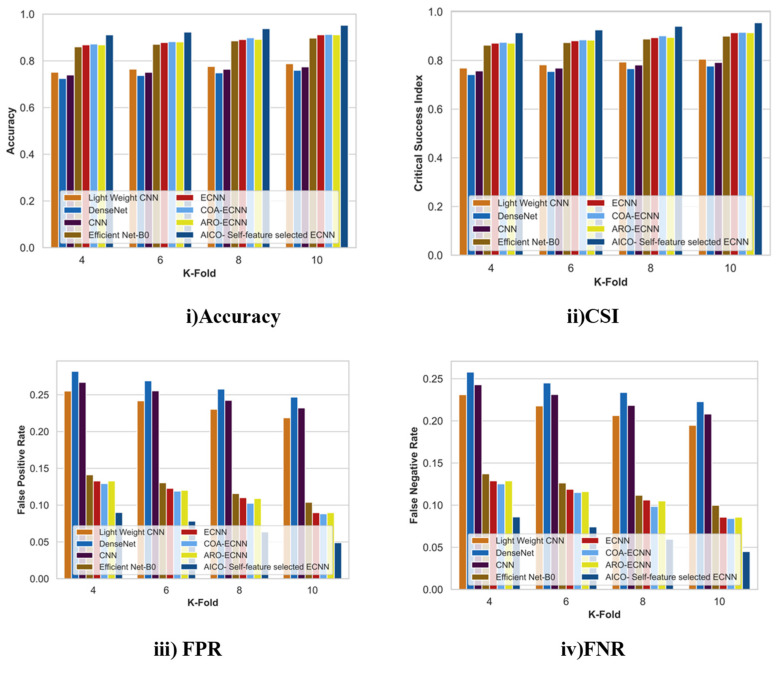
Comparative Analysis with K-fold for the MNIST dataset.

**Figure 10 ijms-25-01546-f010:**
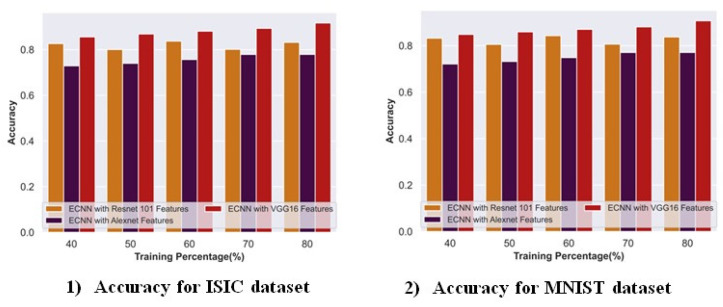
Ablation study on the VGG-16 model with ISIC and the MNIST dataset.

**Figure 11 ijms-25-01546-f011:**
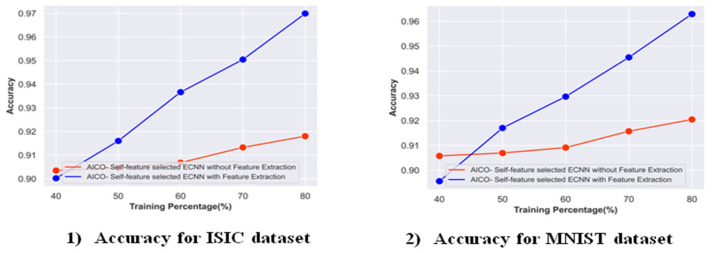
Ablation study on the AICO self-feature selected ECNN with and without feature extraction.

**Figure 12 ijms-25-01546-f012:**
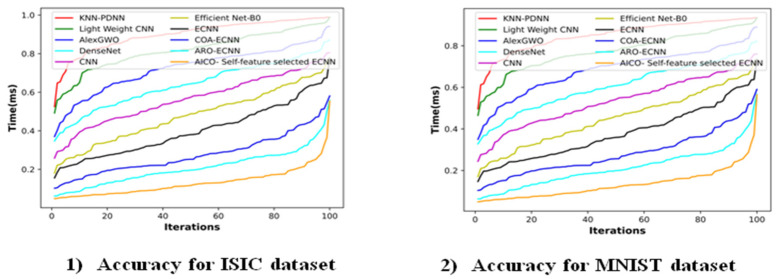
Time complexity analysis on the ISIC and MNIST datasets.

**Figure 13 ijms-25-01546-f013:**
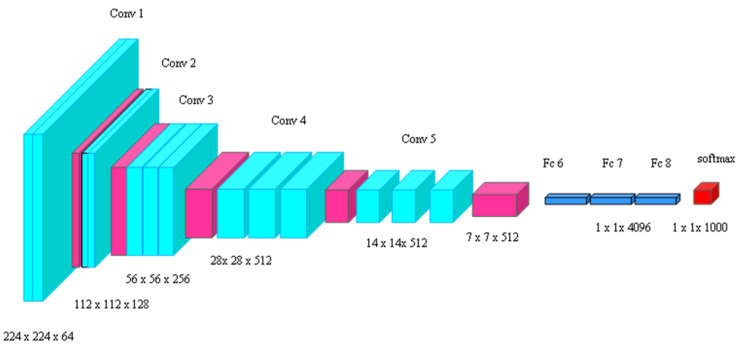
Architecture of VGG 16.

**Figure 14 ijms-25-01546-f014:**
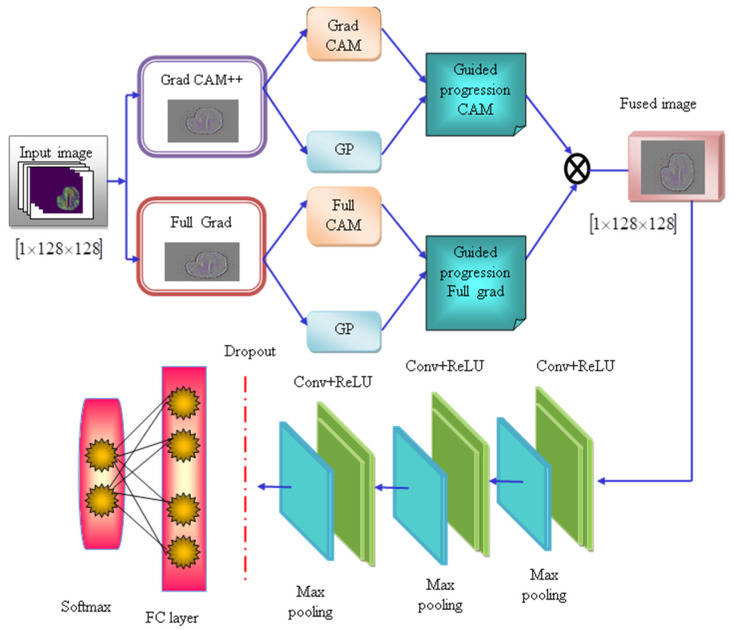
Architecture of the self-feature selected ECNN.

**Figure 15 ijms-25-01546-f015:**
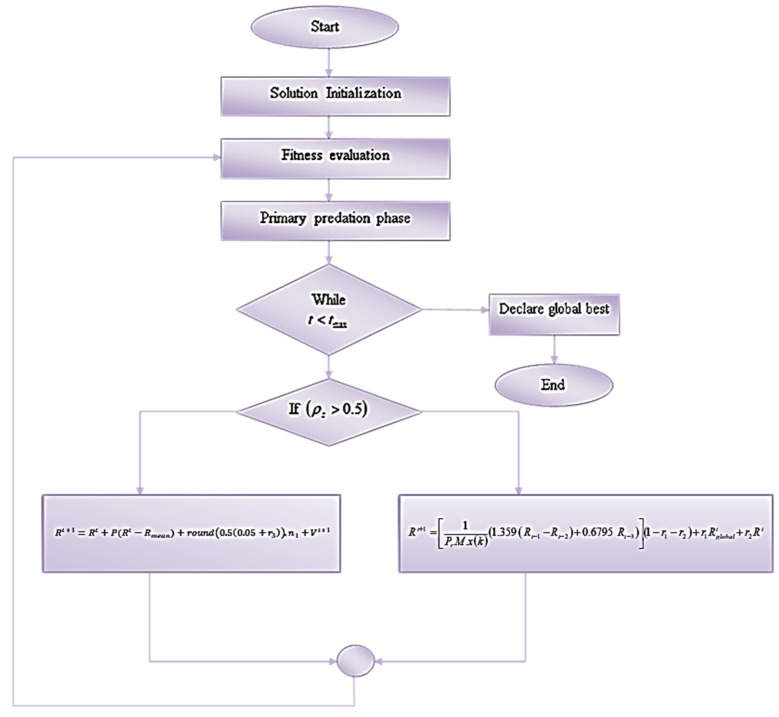
Flowchart of the AICO algorithm.

**Table 1 ijms-25-01546-t001:** Accuracy Comparison of pre-trained models at the Feature Extraction Phase for ISIC and MNIST.

Epochs	ISIC					MNIST				
	ResNet101	AlexNet	MobileNetV2	Xception	VGG16	ResNet101	AlexNet	MobileNetV2	Xception	VGG16
40	0.82	0.73	0.79	0.81	0.86	0.83	0.72	0.81	0.84	0.84
50	0.83	0.74	0.80	0.82	0.87	0.84	0.73	0.82	0.85	0.85
60	0.84	0.76	0.81	0.83	0.88	0.85	0.74	0.82	0.86	0.86
70	0.85	0.78	0.82	0.84	0.89	0.86	0.77	0.83	0.87	0.87
80	0.86	0.80	0.83	0.85	0.92	0.86	0.78	0.84	0.87	0.90
100	0.87	0.81	0.84	0.86	0.95	0.87	0.79	0.84	0.88	0.95

**Table 2 ijms-25-01546-t002:** Performance analysis of TP for the ISIC dataset.

Datasets	Methods	TP 80			
		Accuracy	CSI	FPR	FNR
ISIC	AICO self-feature selected ECNN with Epoch = 20	0.91	0.92	0.09	0.07
	AICO self-feature selected ECNN with Epoch = 40	0.92	0.93	0.07	0.06
	AICO self-feature selected ECNN with Epoch = 60	0.94	0.95	0.06	0.04
	AICO self-feature selected ECNN with Epoch = 80	0.95	0.96	0.05	0.03
	AICO self-feature selected ECNN with Epoch = 100	0.96	0.97	0.03	0.02
Skin Cancer MNIST	AICO self-feature selected ECNN with Epoch = 20	0.90	0.91	0.09	0.08
	AICO self-feature selected ECNN with Epoch = 40	0.92	0.93	0.08	0.06
	AICO self-feature selected ECNN with Epoch = 60	0.93	0.94	0.06	0.05
	AICO self-feature selected ECNN with Epoch = 80	0.95	0.96	0.05	0.03
	AICO self-feature selected ECNN with Epoch = 100	0.96	0.97	0.04	0.02

**Table 3 ijms-25-01546-t003:** A comparative discussion of the AICO self-feature selected ECNN model with existing approaches.

Methods/Analysis			Lightweight CNN	DenseNet	CNN	Efficient Net-B0	ECNN	COA-CAN	ARO-ECNN	AICO Self-Feature Selected ECNN
ISIC dataset	TP 80	Accuracy	0.79	0.76	0.78	0.90	0.91	0.94	0.91	0.96
		CSI	0.81	0.78	0.80	0.91	0.92	0.95	0.92	0.97
		FPR	0.21	0.24	0.22	0.10	0.08	0.05	0.08	0.03
		FNR	0.18	0.21	0.19	0.08	0.07	0.04	0.07	0.02
	K-fold 10	Accuracy	0.78	0.76	0.77	0.90	0.90	0.91	0.90	0.95
		CSI	0.21	0.24	0.22	0.10	0.09	0.08	0.09	0.96
		FPR	0.21	0.24	0.22	0.10	0.09	0.08	0.09	0.04
		FNR	0.19	0.22	0.20	0.09	0.09	0.08	0.09	0.04
MNIST dataset	TP 80	Accuracy	0.79	0.75	0.78	0.90	0.90	0.93	0.92	0.96
		CSI	0.82	0.78	0.80	0.91	0.91	0.94	0.93	0.97
		FPR	0.21	0.25	0.22	0.10	0.09	0.06	0.08	0.04
		FNR	0.17	0.21	0.19	0.08	0.08	0.05	0.06	0.02
	K-fold 10	Accuracy	0.78	0.75	0.77	0.89	0.91	0.91	0.91	0.95
		CSI	0.80	0.77	0.79	0.90	0.91	0.91	0.91	0.95
		FPR	0.21	0.24	0.23	0.10	0.08	0.08	0.08	0.04
		FNR	0.19	0.22	0.20	0.09	0.08	0.08	0.08	0.04

## Data Availability

The experimental datasets used to support this study are publicly available data repositories at https://challenge2020.isic-archive.com (accessed on 10 December 2023).

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
