# Peer review of "Grid-Based Structural and Dimensional Skin Cancer Classification with Self-Featured Optimized Explainable Deep Convolutional Neural Networks"

_ijms, 2024, doi:10.3390/ijms25031546_

Round 1
Reviewer 1 Report
Comments and Suggestions for Authors
see the attachment

Minor editing of English language required
Author Response
Authors Response to Reviewer 1

Reviewer 2 Report
Comments and Suggestions for Authors
Dear Authors,
Thank you for your submission "Grid-Based Structural and Dimensional Skin Cancer Classification with Self-Featured Optimized Explainable Deep Convolutional Neural Networks" to International Journal of Molecular Sciences. The manuscript requires major revision and can be accepted if the following comments can be addressed.
Summary:
The authors propose a grid-based structural and dimensional explainable deep convolution neural network for accurate and interpretable skin cancer classification. The proposed framework involves preprocessing the images, extracting set of features, self-feature selected optimized Explainable CNN optimized using AICO algorithm to detect skin cancer with higher interpretability. The framework leverages on VGG-16 architecture's deep feature extraction capabilities, hierarchical characteristics of skin lesions are extracted from images. The model relies on applying a grid structure over the images to capture spatial relationships within lesions, while dimensional features provide relevant features from image channels. The model has been tested on 2 standard datasets and compared with state of the art skin cancer detection models and proven to have higher accuracy and lesser false positive, false negative rates.
Reviewer Comments:
Major Comments:
1. Paper organization - At the moment, the manuscript provides Experiments, results and discussion before introducing the materials and methods. This really hampers my understanding of the paper. It is necessary to know method used before we discuss the experiments, discussion etc. Please reorganize the sections.
2. Improve Literature Review -
A lot of skin cancer detection techniques are available, for instance, the ones listed below. It is necessary to understand what is the novelty in the proposed approach when compared to the existing ones. Please provide a more comprehensive review of the methods.
a. Nahata, H., Singh, S.P. (2020). Deep Learning Solutions for Skin Cancer Detection and Diagnosis. In: Jain, V., Chatterjee, J. (eds) Machine Learning with Health Care Perspective. Learning and Analytics in Intelligent Systems, vol 13. Springer, Cham. https://doi.org/10.1007/978-3-030-40850-3_8
b. Zghal, Nadia S.; Derbel, Nabil, Melanoma Skin Cancer Detection based on Image Processing, Current Medical Imaging, Volume 16, Number 1, 2020, pp. 50-58(9), https://doi.org/10.2174/1573405614666180911120546
c. V. Srividhya, K. Sujatha, R.S. Ponmagal, G. Durgadevi, L. Madheshwaran, V., Vision based Detection and Categorization of Skin lesions using Deep Learning Neural networks, Procedia Computer Science, Volume 171, 2020, Pages 1726-1735, ISSN 1877-0509, https://doi.org/10.1016/j.procs.2020.04.185.
3. Necessity for each module in the proposed framework -
The authors introduce a framework consisting of modules like preprocessing the images, extracting set of features, self-feature selected optimized Explainable CNN optimized using AICO algorithm.
But there is no information provided why these modules are needed or chosen. It is necessary to provide a rationale behind why each of these modules needs to be included. I would suggest you to bring out limitations in existing techniques and how each of these modules accomplish in achieveing the goal.
4. Preprocessing Stage - The manuscript states that the input images undergo preprocessing and adaptive thresholding ROI extraction. There is no explanation or any paper based on which this is done. Also, what is the effect if the input images are not pre-processed?
5. AICO optimization -
a. Technically, ECNN model should be able to train itself when provided with training data. Why is there a need for optimization?
b. Would it be possible to use any other optimizations and what is the comparison with these methods?
c. If the optimization technique alone can help to improve performance, can we just combine this optimization technique with any other methods.
6. Discussion section is too small saying that the current model doesn't suffer limitations. It would be better if the authors could provide insights on why this doesn't happend for them.
7. Authors mention "Explainability is achieved through attention mechanisms, revealing the basis for the network's decisions.". But in the manuscript, I don't see anything to suggest this is actually happening.
8. Choice of features
a. Why have the authors used these specific features? Can any other features be used?
b. Authors mention they use Grid based for extracting spatial relationship and dimensional features for extracting relevant features. When using Grid based structuring, is there a need for alignment and if so how do you ensure this happens in all cases.
9. Please provide an Ablation study on the features used, modules in framework (optimization method, preprocessing, Adaptive thresholding)?
10. What are the methods ECNN, COA-ECNN, ARO-ECNN? Please provide citation numbers
11. From reference [14], I can see that they achieve a better accuracy on the ISIC dataset, why isn't this included in your comparison with state of the art?
Minor comments to consider-
There needs to be a proper restructuring and English correction for this paper.
1. Line 57 - "they tumor" should be "the tumor".
2. Line 97 - "DensNet" should be "DenseNet"
3. MNIST, MINIST and MINST confusion in usage
There needs to be a proper restructuring and English correction for this paper.
1. Line 57 - "they tumor" should be "the tumor".
2. Line 97 - "DensNet" should be "DenseNet"
3. MNIST, MINIST and MINST confusion in usage
Author Response
Authors response to reviewer 2

Round 2
Reviewer 1 Report
Comments and Suggestions for Authors
Author has incorporated all the comments and manuscript quality has been improved
Comments on the Quality of English LanguageMinor
Reviewer 2 Report
Comments and Suggestions for Authors
Dear Authors,
The manuscript can be accepted at this stage. Thank you for all the changes made.
Thank you